# [^68^Ga]Ga-DFO-c(RGDyK): Synthesis and Evaluation of Its Potential for Tumor Imaging in Mice

**DOI:** 10.3390/ijms22147391

**Published:** 2021-07-09

**Authors:** Sona Krajcovicova, Andrea Daniskova, Katerina Bendova, Zbynek Novy, Miroslav Soural, Milos Petrik

**Affiliations:** 1Department of Organic Chemistry, Faculty of Science, Palacky University, 77900 Olomouc, Czech Republic; sona.krajcovicova@upol.cz; 2Institute of Molecular and Translational Medicine, Faculty of Medicine and Dentistry, Palacky University, 77900 Olomouc, Czech Republic; andrea.daniskova01@upol.cz (A.D.); katerina.bendova01@upol.cz (K.B.); zbynek.novy@upol.cz (Z.N.)

**Keywords:** deferoxamine, RGD peptides, integrins, radiodiagnostics, PET imaging

## Abstract

Angiogenesis has a pivotal role in tumor growth and the metastatic process. Molecular imaging was shown to be useful for imaging of tumor-induced angiogenesis. A great variety of radiolabeled peptides have been developed to target αvβ3 integrin, a target structure involved in the tumor-induced angiogenic process. The presented study aimed to synthesize deferoxamine (DFO)-based c(RGD) peptide conjugate for radiolabeling with gallium-68 and perform its basic preclinical characterization including testing of its tumor-imaging potential. DFO-c(RGDyK) was labeled with gallium-68 with high radiochemical purity. In vitro characterization including stability, partition coefficient, protein binding determination, tumor cell uptake assays, and ex vivo biodistribution as well as PET/CT imaging was performed. [^68^Ga]Ga-DFO-c(RGDyK) showed hydrophilic properties, high stability in PBS and human serum, and specific uptake in U-87 MG and M21 tumor cell lines in vitro and in vivo. We have shown here that [^68^Ga]Ga-DFO-c(RGDyK) can be used for αvβ3 integrin targeting, allowing imaging of tumor-induced angiogenesis by positron emission tomography.

## 1. Introduction

Over the last 30 years, many radiolabeled peptides have been evaluated as promising radiotracers for imaging tumors by means of positron emission tomography (PET) or single photon emission computerized tomography (SPECT) [1,2]. The biological effects of such peptides are mediated via the high affinity targeting of specific receptors. These receptors are often massively overexpressed in numerous cancers, compared to their relatively low density in physiological organs, which is the main principle allowing molecular imaging and therapy of tumors with radiopeptides [3]. Based on the success of studies with ^111^In-labeled somatostatin analogue octreotide (OctreoScan™, Curium, London, UK), the pioneering radiopeptide for tumor imaging, many other receptor-targeting peptides are currently under development or undergoing clinical trials, including arginine-glycine-aspartic acid (RGD)-based peptides [4].

RGD-based family of peptides preferentially bind to the receptors of the integrin superfamily. Integrins are heterodimeric transmembrane receptors interacting with a diverse groups of extracellular ligands [5]. They regulate cellular growth, proliferation, migration, signaling, and cytokine activation and release and thereby play important roles in cell proliferation and migration, apoptosis, tissue repair, as well as in all processes critical to inflammation, infection, and angiogenesis [6]. Among the 24 human integrin subtypes known to date, eight integrin dimers, i.e., αvβ1, αvβ3, αvβ5, αvβ6, αvβ8, α5β1, α8β1, and αIIbβ3, recognize the tripeptide RGD motif within extracellular matrix proteins and represent the most important integrin receptor subfamily involved in cancer and the metastatic process [7]. Of these, the integrin αvβ3 has been studied most extensively for its role in tumor angiogenesis using nuclear medicine imaging techniques [8]. A large variety of radiotracers based on RGD peptides have been developed and tested for targeting αvβ3 integrin in both preclinical and clinical settings.

Integrin-specific radiotracers are obtained by radiolabeling of RGD-based precursors, which can be used for imaging with scintigraphy and SPECT (by using gamma emitters like, e.g., Tc-99m and In-111) or PET (by using positron-emitting radionuclides, e.g., F-18, Ga-68, or Cu-64). For labeling of peptide-precursors with the use of a radiometal (e.g., Ga-68, Cu-64, Tc-99m, In-111), a specific metal chelating agent has to be introduced to the precursor’s structure [9]. The choice of the chelating agent is largely determined by the nature and oxidation state of the radiometal to be used for labeling [10]. Among various chelating agents for radiometal labeling of RGD peptides, 1,4,7,10-tetraazacyclododecane-1,4,7,10-tetraacetic acid (DOTA), 1,4,7-triazacyclononane-1,4,7-triacetic acid (NOTA) and their derivatives are the most widely used [11].

Many previous works have shown that deferoxamine (DFO), a hexadentate hydroxamate siderophore, is a common and suitable chelator for labeling with radiometals such as Ga-67, Zr-89, In-111, and Ga-68 [12,13,14,15,16]. Although a series of near-infrared fluorescent conjugates containing DFO and multi-RGD peptide moieties were designed, synthesized, and affinity to αvβ3 integrin was evaluated in vitro [17], radiolabeled DFO-RGD peptide conjugates were not studied neither in vitro nor in vivo. Herein, we report the preparation of cyclic c(RGDyK) pentapeptide conjugated with *p*-SCN-Bn derivatized DFO to obtain the DFO-based c(RGDyK) conjugate, which was then labeled with Ga-68. Binding properties of [^68^Ga]Ga-DFO-c(RGDyK) towards αvβ3 integrin were studied in vitro and in vivo, including PET/CT imaging in a mouse tumor model.

## 2. Results and Discussion

### 2.1. Conjugate Synthesis

To improve the pharmacokinetic profile, especially in vivo stability against enzymatic cleavage of RGD peptides alongside maintenance of high activity and specificity for αvβ3 integrin, several approaches have been developed in the past [18]. The enrichment of the pentapeptide chain with amino acids in unnatural *D*-configuration and subsequent cyclization were demonstrated to be significant in vivo stability improvements and are routinely used in the development of RGD peptides nowadays [19]. In our study, the c(RGDyK) sequence for conjugation with DFO chelating agent was chosen, as it is one of the most prominent structures for the development of molecular imaging compounds in order to determine αvβ3 expression. It was extensively studied both preclinically and clinically with different radiometal chelators for tumor imaging [20,21,22,23,24,25,26]. Although DFO is well-accessible and an established chelating agent in nuclear medicine for peptide and particularly antibody radiolabeling, to our knowledge, DFO-based RGD peptides for radiometal labeling have not been studied yet. DFO is a molecule known for its ability to bind many different metal ions for years [27]. In 1968, it was also approved for medical use by the FDA under the brand name Desferal^®^ (Novartis, Basel, Switzerland) and became a well-established clinically used medication [16]. In nuclear medicine, DFO-based compounds offer possibilities for labeling with different radiometals as mentioned above, allowing a wide range of diagnostic as well as therapeutic applications [12,15,17].

The proposed synthetic pathway started from the commercially available 2-chlorotrityl chloride polystyrene resin **1**, which was acylated with Fmoc-Gly-OH in the presence of *N,N’*-diisopropylethylamine (DIPEA) as a base (Scheme 1) to give **2**. For the cleavage of Fmoc protecting group, a non-nucleophilic strong base 1,8-diazabicyclo[5.4.0]undec-7-ene (DBU) in CH_2_Cl_2_ was used, as we observed lower crude purities within the reaction sequence when the traditional cleavage protocol with piperidine in dimethylformamide (DMF) was applied. Construction of oligopeptide was accomplished by the conventional solid-phase peptide synthesis with *N*,*N*′-diisopropylcarbodiimide (DIC) and hydroxybenzotriazole (HOBt) as the activating agents. Notably, we decided to incorporate *D*-tyrosine (y) amino acid into the linear sequence that should result in enhanced stability of the RGD peptide, as was mentioned above. The corresponding intermediates **3**–**6** were obtained in high crude purities (according to spectral data, see Appendix A). The loading of pentapeptide **6** was quantified to 0.7 mmol/g. It is worth mentioning that the 4,4-dimethyl-2,6-dioxocyclohex-1-ylidene (Dde) protecting group of the lysine side chain was chosen due to its orthogonality with acid-labile protecting groups (Pbf, *tert*-butyl) as well as the Fmoc protecting group [28].

Prior to the cyclization step, the linear peptide **6** was liberated from the resin (Scheme 2). Chemoselective cleavage (to maintain residual protecting groups) was performed with hexafluoroisopropanol (HFiP) in CH_2_Cl_2_. Following cyclization of **7** using benzotriazol-1-yl-oxytripyrrolidinophosphonium hexafluorophosphate (PyBOP) yielded the protected cyclized peptide **8**. The Dde protecting group was then cleaved with hydrazine (2%) which furnished the key intermediate **8** in excellent crude purity applicable for further modification with DFO (Scheme 2).

For the attachment of DFO, commercially available *p*-SCN-Bn-deferoxamine was applied. The reaction proceeded smoothly and with high crude purity of the corresponding intermediate. Following acid-mediated cleavage of residual protecting groups in trifuoroacetic acid (TFA) yielded the final DFO-based c(RGDyK) conjugate **9** (Scheme 3), which was purified using semipreparative reversed-phase high-performance liquid chromatography (RP-HPLC) and fully characterized (see Appendix A).

### 2.2. Radiolabeling and In Vitro Characterization

DFO-c(RGDyK) **9** was radiolabeled with gallium-68 with molar activity of up to 6 GBq/μmol and radiochemical purity >98% in 5 min at 85 °C, confirmed by RP-HPLC (corresponding radiochromatogram is shown in Figure 1). The ^68^Ga-labeled tracer was used without further purification for all the experiments. Gallium-68 is a positron emitter that decays with a half-life of 67.7 min and positron branching of 89.1%, emitting high-energy positrons of ca. 1.9 MeV [29,30]. The relatively short half-life can be a limitation, therefore, the short reaction time and high radiochemical yield and/or purity preventing any further purification or post-processing steps are important factors for the preparation of ^68^Ga-labeled radiopharmaceuticals. In recent years, gallium-68 has attracted increasing interest in the field of nuclear medicine, currently being most often utilized in radiopharmaceuticals for oncology diagnostics [31].

[^68^Ga]Ga-DFO-c(RGDyK) showed hydrophilic properties (log P = −2.01 ± 0.08) with plasma protein binding approximately 30% after 120 min of incubation at 37 °C in human serum. The in vitro stability of [^68^Ga]Ga-DFO-c(RGDyK) was high in human serum and PBS (>97% in all tested time points), while in the presence of high excess of a competing metal and chelator, the stability of [^68^Ga]Ga-DFO-c(RGDyK) decreased rapidly (see Table 1). The obtained in vitro data are in a good agreement with previously published data on [^68^Ga]Ga-DFO [16]. However, [^68^Ga]Ga-DFO-c(RGDyK) revealed higher lipophilicity, plasma protein binding and instability in the presence of a competing metal and chelator as compared with those of [^68^Ga]Ga-NODAGA-c(RGDyK) [24], which subsequently influenced the in vivo behavior of the studied radiotracer.

In vitro uptake assays in tumor cell lines (U-87 MG, M21 and M21-L) showed specific uptake of [^68^Ga]Ga-DFO-c(RGDyK) by cell lines expressing αvβ3 integrin (U-87 MG and M21), which could be blocked with an excess of appropriate competing inhibitor (NODAGA-c(RGDyK)). In the uptake assay using U-87 MG cells, the uptake of [^68^Ga]Ga-DFO-c(RGDyK) increased with the incubation time and could be blocked with NODAGA-c(RGDyK) by about half (Figure 2A). This is consistent with the findings of Novy et al. [24], who showed similar in vitro uptake behavior of [^68^Ga]Ga-NODAGA-c(RGDyK) in a U-87 MG cell line. In the cell uptake assay using M21 and M21-L cells, the uptake of [^68^Ga]Ga-DFO-c(RGDyK) could only be blocked for the αvβ3-positive cells (M21), whereas for the αvβ3-negative control cells (M21-L), the uptake was approximately one half of the amount for the receptor-positive cells. The binding of [^68^Ga]Ga-DFO-c(RGDyK) to the M21-L cells was very similar under blocked and unblocked conditions (Figure 2B). These results are in accordance with the data published by Knetsch et al. [32], who examined the in vitro uptake of [^68^Ga]Ga-NODAGA-c(RGDfK) in M21 and M21-L cell lines. It also confirms that the small change in the peptide sequence from c(RGDyK) to c(RGDfK) does not have any significant impact on the αvβ3 integrin binding affinity [19].

The in vitro competition assays using increasing amounts of cold DFO-c(RGDyK) **9** showed that the inhibitory peptide was able to suppress the binding of [^68^Ga]Ga-NODAGA-c(RGDyK) to the αvβ3 integrin expressing M21 cells and that the binding kinetics followed a classic sigmoid inhibition curve. The IC_50_ value found for DFO-c(RGDyK) **9** was 2.35 ± 1.48 nM (Figure 2C). Although we did not use the gold standard radioligand [^125^I]I-echistatin and observed relatively high nonspecific binding in the competition assay, which could affect the accuracy of IC_50_ calculations, the determined IC_50_ value is in accordance with the IC_50_ values of similar RGD-based conjugates published by Kapp et al. [19]. Other than monomeric RGD-based peptides, multimeric compounds presenting more than one RGD motif have also been introduced [18]. This “multimerisation” approach may result in improved target affinity and prolonged target retention, however, it may have a significant, and in some cases undesired, impact on in vivo behavior of the RGD conjugates [18,33].

### 2.3. In Vivo Characterization

Biodistribution studies were performed 30 and 90 min after [^68^Ga]Ga-DFO-c(RGDyK) injection in healthy Balb/c and tumor-bearing (U-87 MG) Balb/c nude mice. [^68^Ga]Ga-DFO-c(RGDyK) displayed relatively rapid excretion mainly via the renal system and showed minimal retention in blood and other organs in healthy animals 90 min after injection (Figure 3). The highest activity concentration in the organs at a later time point (90 min p.i.) was found in the kidneys (5.31 ± 0.12% ID/g), intestines (1.99 ± 0.01% ID/g), and the stomach (1.78 ± 0.05% ID/g) in healthy animals. The ex vivo biodistribution data were consistent with the results obtained from PET/CT imaging 30 and 90 min p.i. (Figure 3).

The results of biodistribution studies of [^68^Ga]Ga-DFO-c(RGDyK) in U-87 MG xenografted Balb/c nude mice are presented in Figure 4. [^68^Ga]Ga-DFO-c(RGDyK) in tumor-bearing animals showed in vivo behavior similar to that in healthy mice. In addition, [^68^Ga]Ga-DFO-c(RGDyK) was accumulated in tumor tissue 30 min p.i. (3.03 ± 0.62% ID/g), and tumor washout was rather slow, as a significant amount of radioactivity (1.54 ± 0.56% ID/g), compared to that in other organs, was still present in tumor tissue after 90 min. In general, this is in good agreement with reports on other radiolabeled RGD peptides being evaluated on the similar integrin αvβ3-expressing tumor model [20,24,33,34]. [^68^Ga]Ga-DFO-c(RGDyK) displayed lower U-87 MG tumor uptake in vivo compared to that of multimeric RGD peptide counterparts and confirmed that multimerization usually improves tumor-targeting capability of radiolabeled peptides [18,35], including RDG-based peptides [36,37]. However, compared to monomeric conjugates, e.g., [^68^Ga]Ga-DOTA-c(RGDyK) [23], [^68^Ga]Ga-NOTA-c(RGDyK) [20], and [^68^Ga]Ga-NODAGA-c(RGDyK) [24], [^68^Ga]Ga-DFO-c(RGDyK) showed slightly slower pharmacokinetics and higher U-87 tumor accumulation and retention, which is in full agreement with the obtained in vitro data. Considering not only tumor uptake but also the pharmacokinetics and tissue distribution, monomeric or dimeric RGD-based peptide conjugates seem to be the most promising candidates for in vivo imaging of integrins-expressing tumors [20,33,37].

Animal PET/CT imaging studies were conducted in healthy mice and in the same tumor mouse model (U-87 MG) as biodistribution studies. Moreover, PET/CT imaging was also performed in M21(αvβ3-positive) and M21-L (αvβ3-negative) xenografted Balb/c nude mice to evaluate the specificity of [^68^Ga]Ga-DFO-c(RGDyK) uptake in the tumors with high and low αvβ3 integrin expression in vivo. Static PET/CT images of [^68^Ga]Ga-DFO-c(RGDyK) in U-87 MG tumor mice (Figure 5A) confirmed the results from ex vivo biodistribution showing the uptake in αvβ3 integrin-overexpressing tissue and clearly visualizing the tumor 30 and 90 min p.i. with relatively decent target-to-organ contrast observed for both time points. PET/CT imaging of [^68^Ga]Ga-DFO-c(RGDyK) in M21 and M21-L tumor-bearing mice (Figure 5B) displayed specific uptake of the radiotracer in αvβ3 integrin-positive (M21) tumor tissue, while no uptake was observed in αvβ3 integrin-negative (M21-L) tumor tissue both 30 and 90 min p.i. Animal PET/CT imaging of [^68^Ga]Ga-DFO-c(RGDyK) in both tumor mouse models (U-87 MG and M21) confirmed that it has similar in vivo behavior as other analogical radiolabeled RGD-based peptides [20,23,24] and can be used for PET imaging of tumor angiogenesis. Moreover, DFO allows labeling with different radionuclides [12,15,38,39], which could open ways of the studied DFO-c(RGDyK) for SPECT imaging and theranostic applications [18,40].

## 3. Materials and Methods

### 3.1. Chemicals

All reagents were purchased from commercial sources as reagent or analytical grade and used without further purification. Solvents and chemicals were purchased from Sigma-Aldrich (St. Louis, MO, USA), Acros Organics (Geel, Belgium), AAPPTec (Louisville, KY, USA), or Fluorochem (Hadfield, UK). *p*-SCN-Bn-Deferoxamine was purchased from Macrocyclics (Plano, TX, USA). Anhydrous solvents were dried over 4 Å molecular sieves or stored as received from commercial suppliers. ^68^GaCl_3_ for radiolabeling was eluted from a ^68^Ge/^68^Ga-generator (Eckert & Ziegler Eurotope GmbH, Berlin, Germany) with 0.1 N HCl using a fractionated elution approach.

### 3.2. Conjugate Preparation

Reactions were performed in plastic reaction vessels (syringes, each equipped with a porous disk) using a manually operated synthesizer (Torviq, Tuscon, AZ, USA) or in ace-pressure tubes, unless stated otherwise. The volume of wash solvent was 10 mL per 1 g of resin. For washing, resin slurry was shaken with the fresh solvent for at least 1 min before changing the solvent. Resin-bound intermediates were dried under a stream of nitrogen for prolonged storage and/or quantitative analysis.

#### 3.2.1. Procedure for Acylation with Fmoc-Gly-OH

2-Chlorotrityl chloride resin **1** (300 mg, loading 0.85 mmol/g) was added to the polypropylene fritted syringe and then washed with CH_2_Cl_2_ (3 × 5 mL). The solution of Fmoc-Gly-OH (267 mg, 0.9 mmol) and DIPEA (148 μL, 0.9 mmol) in a mixture of DMF/CH_2_Cl_2_ (1:1, 3 mL, *v*/*v*) was added to resin **1**. The reaction slurry was shaken at ambient temperature for 16 h, followed by washing with CH_2_Cl_2_ (5 × 5 mL), DMF (5 × 5 mL), and CH_2_Cl_2_ (5 × 5 mL). For capping, the solution of CH_2_Cl_2_/methanol/DIPEA (17:2:1, 10 mL, *v*/*v*) was added to the resin, and the slurry was shaken for an additional 1 h. Then, the resin was washed again with CH_2_Cl_2_ (3 × 5 mL). Subsequent cleavage from the resin (according to General procedure C) confirmed full conversion to product **2**.

Loading after this step was determined as follows: the sample of resin **2** (~30 mg) was washed with CH_2_Cl_2_ (5 × 3 mL) and MeOH (3 × 3 mL), dried under a stream of nitrogen and divided into two portions (2 × 12 mg). Both samples were treated with CH_2_Cl_2_/TFA (1:1, 1 mL, *v*/*v*) for 1 h, after which the cleavage cocktail was evaporated under a stream of nitrogen. Cleaved compounds were dissolved in CH_3_CN/H_2_O (1:1, 1 mL, *v*/*v*), diluted four times, and analyzed by ultra-high performance liquid chromatography coupled with mass spectrometry and ultraviolet detection (UHPLC/MS/UV). Loading of the resin was calculated with the use of an external standard (Fmoc-Ala-OH, 0.5 mg/mL).

#### 3.2.2. General Procedure A for Deprotection of Fmoc

Corresponding resin (300 mg) was swollen in CH_2_Cl_2_ (5 mL) for 30 min. Solution of DBU/CH_2_Cl_2_ (1:1, 5 mL, *v*/*v*) was added to the resin, and the reaction slurry was shaken for an additional 10 min, then washed again with CH_2_Cl_2_ (3 × 5 mL) and DMF (3 × 5 mL). The resin was used in the next step without further analysis.

#### 3.2.3. General Procedure B for Preparation of Pentapeptide

The corresponding resin (300 mg) was swollen in CH_2_Cl_2_ (5 mL) for 30 min and then washed with CH_2_Cl_2_ (3 × 5 mL) and DMF (3 × 5 mL). The Fmoc-protected amino acid (0.2 M) and HOBt (91 mg, 0.6 mmol) were dissolved in DMF (3 mL), and DIC (94 μL, 0.6 mmol) was added. The reaction mixture was added to the corresponding Fmoc-deprotected resin (according to General procedure A) and shaken for 4–16 h, followed by washing with CH_2_Cl_2_ (5 × 5 mL), DMF (5 × 5 mL), and CH_2_Cl_2_ (5 × 5 mL). Subsequent cleavage from the resin (according to General procedure C) confirmed the full conversion to products **3**–**6**. The loading of **6** was determined according to the above-mentioned procedure.

#### 3.2.4. Procedure for HFiP Mediated Cleavage from the Resin (with Maintaining Protecting Groups)

Resin **6** (300 mg) was swollen in CH_2_Cl_2_ (5 mL) for 30 min and then washed with CH_2_Cl_2_ (3 × 5 mL) and DMF (3 × 5 mL). Fmoc protecting group was cleaved according to General procedure A. The solution of 1,1,1,3,3,3-hexafluoro-2-propanol (HFiP) in CH_2_Cl_2_ (1:4, 3 mL, *v*/*v*) was added to the resin, and the reaction slurry was shaken at ambient temperature for 3 h. Then, the resin was washed with a HFiP/CH_2_Cl_2_ (1:4, 3 × 4 mL, *v*/*v*) mixture, organic extracts were combined, and residual solvents were evaporated under reduced pressure. UHPLC/MS analysis confirmed quantitative cleavage of **6** from the resin, and the crude product **7** was used directly in the next step without further purification.

#### 3.2.5. Procedure for Peptide Cyclization

Crude product **7** was dissolved in DMF (3 mL/300 mg of **6**) with subsequent addition of PyBOP (156 mg, 0.3 mmol) and DIPEA (200 μL, 1.2 mmol). The reaction mixture was stirred at ambient temperature for 24 h. UHPLC/MS analysis confirmed formation of the cyclized product (ESI−: 1147) which was used directly in the next step.

#### 3.2.6. Procedure for Deprotection of Dde

NH_2_NH_2_·OH (60 μL) was added into the reaction mixture with cyclized peptide, and the reaction mixture was stirred for an additional 3 h. Subsequent UHPLC/MS analysis confirmed the formation of **8**. The residual solvents were evaporated on high vacuum, and crude product **8** was purified by semipreparative HPLC.

#### 3.2.7. Procedure for Acylation with *p*-SCN-Bn-Deferoxamine and Final Deprotection of Pbf and Tert-Butyl Protecting Groups

Compound **8** (15 mg, 0.01 mmol) was dissolved in anhydrous DMF (500 μL). The solution of *p*-SCN-Bn-deferoxamine (12 mg, 0.01 mmol) and DIPEA (10 μL, 0.05 mmol) in anhydrous dimethyl sulfoxide (DMSO; 500 μL) was added, and the resulting mixture was stirred at ambient temperature for 1 h, after which UHPLC/MS analysis confirmed the formation of conjugate. The residual solvents were evaporated under high vacuum, the solution of CH_2_Cl_2_/TFA (1:1, 1 mL, *v*/*v*) was added, and the reaction mixture was stirred at ambient temperature for an additional 2 h. Residual solvents were evaporated in a stream of nitrogen, and the final product **9** was purified by semipreparative HPLC.

#### 3.2.8. Instrumentation and Analytics

For the UHPLC-MS analysis, a sample of resin (~5 mg) was treated with CH_2_Cl_2_/TFA (1:1, 1 mL, *v*/*v*), the cleavage cocktail was evaporated under a stream of nitrogen, and cleaved compounds extracted into CH_3_CN/H_2_O (1:1, 1 mL, *v*/*v*). Prior to HPLC separation (column Phenomenex Gemini, 50 × 2.00 mm, 3 µm particles, C18), the samples were injected by direct infusion into the mass spectrometer using an autosampler. Mobile phase was isocratic 80% CH_3_CN and 20% 0.01 M ammonium acetate in H_2_O or 95% methanol + 5% H_2_O + 0.1% formic acid and flow of 0.3 mL/min.

Liquid chromatography coupled with mass spectrometry (LC/MS) analyses were carried out on a UHPLC-MS system consisting of a UHPLC chromatograph Acquity with a photodiode array detector and a single quadrupole mass spectrometer (Waters, Milford, MA, USA), using X-Select C18 column at 30 °C and flow rate of 0.6 mL/min. Mobile phase was (A) 0.01 M ammonium acetate in H_2_O and (B) CH_3_CN, linearly programmed from 10% A to 80% B over 2.5 min, kept for 1.5 min. The column was re-equilibrated with 10% of solution B for 1 min. The ESI source operated at a discharge current of 5 μA, vaporizer temperature of 350 °C, and capillary temperature of 200 °C. HPLC purification was carried out on a C18 reverse phase column (YMC Pack ODS-A (YMC, Kyoto, Japan), 20 × 100 mm, 5 μm particles), gradient was formed from CH_3_CN and 0.01 M ammonium acetate in H_2_O, flow rate of 15 mL/min. For lyophilization of residual solvents at −110 °C, a ScanVac Coolsafe 110-4 (Labogene, Lillerod, Denmark) was used.

High-resolution mass spectrometry (HRMS) analyses were performed using an LC chromatograph (Dionex Ultimate 3000, Thermo Fischer Scientific, Waltham, MA, USA) and an Exactive Plus Orbitrap high-resolution mass spectrometer (Thermo Fischer Scientific, Waltham, MA, USA) operating at positive full scan mode (120,000 FWMH) in the range of 100–1000 *m*/*z*. The settings for electrospray ionization were as follows: oven temperature of 150 °C and source voltage of 3.6 kV. The acquired data were internally calibrated with phthalate as a contaminant in methanol (*m*/*z* 297.15909). The samples were diluted to a final concentration of 0.1 mg/mL in CH_3_CN/H_2_O (9:1, *v*/*v*).

Nuclear magnetic resonance (NMR) spectra were recorded on a JEOL ECX500 spectrometer (JEOL, Tokyo, Japan) at magnetic field strengths of 11.75 T with operating frequencies 500.16 MHz (for ^1^H), and 125.77 MHz (for ^13^C) at 27 °C. Chemical shifts (δ) are reported in parts per million (ppm) and coupling constants (J) are reported in Hertz (Hz). The ^1^H and ^13^C NMR chemical shifts (δ in ppm) were referenced to the residual signals of DMSO-*d_6_* [2.50 (^1^H) and 39.52 (^13^C)]. The residual signal of ammonium acetate (from HPLC purification) exhibited a signal at 1.90 ppm (^1^H) and at 21.3 ppm and 172.0 ppm (^13^C).

Abbreviations in NMR spectra: br s—broad singlet, d—doublet, dd—doublet of doublets, m—multiplet, s—singlet.

### 3.3. Radiolabeling and Quality Control

For radiolabeling of DFO-c(RGDyK) **9**, ^68^GaCl_3_ was obtained by eluting a commercial ^68^Ge/^68^Ga-generator with 0.1 N HCl. The fractionated elution method was used to increase the radioactivity to volume ratio to its maximum. DFO-c(RGDyK) **9** (0.1–50 μg) was incubated with 300 μL of ^68^Ge/^68^Ga-generator eluate (20–40 MBq), and pH was adjusted to 4–5 by adding 30 μL of 1.14 M CH_3_COONa × 3H_2_O. After the incubation (1–30 min) at 85 °C and the adjustment of pH to 6–7 with 100 μL of 1.14 M CH_3_COONa × 3H_2_O, the samples were analyzed by RP-HPLC with radiodetection.

Radiochemical purity (RCP) of [^68^Ga]Ga-DFO-c(RGDyK) was determined by RP-HPLC using the gradient method. RP-HPLC analysis was performed with a Dionex UltiMate 3000 UHPLC system (Thermo Fisher Scientific, Waltham, MA, USA) consisting of an UltiMate 3000 RS pump, an UltiMate 3000 autosampler, an Ultimate 3000 column compartment (25 °C oven temperature), an UltiMate 3000 variable wavelength detector (UV detection at λ = 220 and 280 nm), and a GABI Star radiometric detector (Raytest GmbH, Straubenhardt, Germany). A Nucleosil 120-5 C18 250 × 40 mm column (WATREX, Prague, Czech Republic) with 1 mL/min flow rate was used with the following gradient: CH_3_CN/H_2_O/0.1% trifluoroacetic acid (TFA): 0–3.0 min 0% CH_3_CN, 3.1–6.0 min 0–50% CH_3_CN, 6.1–10.0 min 50% CH_3_CN, 10.1–13.0 min 80% CH_3_CN, 13.1–15 min 0% CH_3_CN.

### 3.4. Partition Coefficient, In Vitro Stability, and Protein Binding

Partition coefficient (log P) of [^68^Ga]Ga-DFO-c(RGDyK) was determined as follows: Radiolabeled DFO-c(RGDyK) was dissolved with phosphate-buffered saline (PBS) pH = 7.4 to 1 mL (~7 μM). Aliquots of 50 μL were added to 450 μL PBS and 500 μL octanol, and the mixture was vigorously vortexed for 20 min at 1500 rpm. The aqueous and organic solvents were separated by centrifugation (2 min at 2000× *g*), and 50 μL aliquots of both layers were collected and measured in a gamma counter (PerkinElmer, Waltham, MA, USA). Log *p* values were calculated from the obtained data in Microsoft Office Excel 2010 (Microsoft, Redmond WA, USA) (mean of n = 6).

In vitro stability of [^68^Ga]Ga-DFO-c(RGDyK) was studied in different media, including PBS, human serum (Sigma Aldrich, St. Louis, MO, USA), competing metal (0.1 M FeCl_3_), and chelator solution (6 mM diethylenetriaminepentaacetic acid (DTPA)). [^68^Ga]Ga-DFO-c(RGDyK) was directly mixed with PBS and human serum at a 1:10 ratio, and with FeCl_3_ and DTPA solutions at a 1:1 ratio. Thus, the prepared reaction mixtures were incubated for 30, 60, and 120 min, respectively, at 37 °C. After incubation, human serum samples were precipitated with acetonitrile and centrifuged (3 min, 2000× *g*). The supernatant was analyzed by RP-HPLC. Samples containing PBS, DTPA, and FeCl_3_ were analyzed directly. The stability is reported as % RCP of [^68^Ga]Ga-DFO-c(RGDyK) (n = 3).

For protein binding measurement, 10 μL of [^68^Ga]Ga-DFO-c(RGDyK) was mixed with 190 μL of human serum or PBS as a control and incubated at 37 °C up to 120 min. The samples were collected at selected time points (30, 60, and 120 min after incubation) and analyzed by size exclusion chromatography using MicroSpin G-50 Columns (GE Healthcare, Buckinghamshire, UK) in triplicates. First, the columns were centrifuged at 2000× *g* for 1 min to remove the storage buffer. After adding 25 μL of tested sample, the columns were centrifuged again for 2 min with 2000× *g*. Column and eluate were measured in the gamma counter, and percentages of non-protein-bound radiolabeled peptide (column) and protein-bound radiotracer (eluate) were calculated.

### 3.5. Cell Culture and In Vitro Cell Assays

Human glioblastoma multiforme U-87 MG cells (ATCC, Manassas, VA, USA) were cultured in Dulbecco’s Modified Eagle Medium (Merck, Darmstadt, Germany) supplemented with 10% fetal bovine serum, 0.1 mM non-essential amino acids, and 1.0 mM sodium pyruvate at 37 °C in a 5% carbon dioxide humidified incubator. Human melanoma cell lines M21 (fluorescence-activated cell sorting selected clone, which stably express αvβ3 integrin; αvβ3 integrin-positive) and M21-L (αvβ3 integrin-negative) originated from prof. Cheresh, Departments of Immunology and Vascular Biology, The Scripps Research Institute, La Jolla, CA, USA and were a kind gift of prof. Decristoforo, Department of Nuclear Medicine, Medical University Innsbruck, Austria. Both M21 and M21-L cells were cultured in Dulbecco’s Modified Eagle Medium (Merck, Darmstadt, Germany) supplemented with 10% fetal bovine serum. All the cells were subcultured and used for experiments at a confluency of 70–90%.

Cell uptake assays were performed in 24-well plates cell at cellular confluency of 70–90%. In the case of U-87 MG, collagen-coated plates (Waltham, MA, USA) were employed due to the insufficient adherence of this cell line. The uptake assay itself was carried out using a dedicated buffer instead of the regular cell culture media. This buffer consisted of 25 mM Tris/HCl, 5.4 mM KCl, 1.8 mM CaCl_2_, 0.8 mM MgSO_4_, 5 mM glucose, and 140 mM NaCl in H_2_O [41]. The cells were incubated with [^68^Ga]Ga-DFO-c(RGDyK) (7 nM) alone and also with an excess (3.5 µM) of cold NODAGA-c(RGDyK) (ABX, Radeberg, Germany) as the uptake inhibitor. This incubation was performed in the buffer described above. The incubation times with ^68^Ga-labeled ligand were 30–90 min in the case of U-87 MG cells and 60 min in the case of M21 and M21-L cells at 37 °C. Afterwards, the 24-well plates were rinsed with PBS and the cells were lysed by 0.1 M NaOH and measured for radioactivity in the gamma counter. The uptake of [^68^Ga]Ga-DFO-c(RGDyK) was calculated as the mean of the percentage of the applied dose ± standard deviation.

The determination of the IC_50_ value of [^68^Ga]Ga-DFO-c(RGDyK) in vitro was carried out using an M21 cell line. This competition assay was performed in standard 24-well plates with the cells seeded one day before the experiment. The assay itself employed the same incubation buffer as in the above-described uptake assay. Briefly, the M21 cells were washed with PBS buffer, and cold DFO-c(RGDyK) **9** was added (triplicate wells) in nine different concentrations to cover a concentration range from 1 × 10^−10^ to 5 × 10^−5^ M. The cells were then incubated for 30 min at 37 °C. Next, [^68^Ga]Ga-NODAGA-c(RGDyK) (15 nM) was added, followed by 60 min of incubation at 37 °C. Thereafter, the uptake was interrupted by buffer removal. The cells were rinsed with ice-cold PBS buffer and lysed with 0.1 M NaOH. The lysed cell were collected for radioactivity and protein content measurement. The protein content was determined using a standard BCA protein assay (Pierce™ BCA Protein Assay Kit, Thermo Fisher Scientific, Waltham, MA, USA). Radioactivity of the cell samples was measured in the gamma counter.

The cellular uptake of [^68^Ga]Ga-NODAGA-c(RGDyK) was calculated as the percentage of applied dose per milligram of cell protein. These uptake values were used to plot a classical sigmoidal dose-response curve in GraphPad Prism (GraphPad Software, San Diego, CA, USA). The IC_50_ value was determined from this curve using fitting analysis in the above-mentioned software.

### 3.6. Animal Experiments

The animal studies were performed using female 8–10-week-old Balb/c and athymic Balb/c nude mice (Envigo, Horst, The Netherlands). The animals were acclimatized to laboratory conditions for 1 week prior to experimental use and housed in a specific-pathogen-free animal facility with free access to animal chow and water. During the experiments, general health and body weight of the animals were monitored. The number of animals was reduced as much as possible (n = 3 per group and time point) for all in vivo experiments. The tracer injection as well as small animal imaging was carried out under 2% isoflurane anesthesia (FORANE, Abbott Laboratories, Abbott Park, IL, USA) to minimize animal suffering and to prevent animal motion.

For animal tumor models, athymic Balb/c nude mice were subcutaneously injected in the right flank with 5 × 10^6^ U-87 MG cells mixed with Matrigel Matrix (Corning, NY, USA) at a 1:1 ratio or in the right and left flank with 5 × 10^6^ M21 and M21-L cells. Tumor growth was continuously monitored by caliperation. When the tumor volume reached around 0.1–0.3 cm^3^ (i.e., 6–8 weeks after the inoculation of tumor cells), the mice were used for ex vivo biodistribution studies or PET/CT imaging.

To evaluate pharmacokinetics and biodistribution of [^68^Ga]Ga-DFO-c(RGDyK) in healthy and tumor-bearing (U-87 MG cell line) animals ex vivo, a group of three Balb/c or athymic Balb/c nude mice per time point were retro-orbitally (r.o.) injected with [^68^Ga]Ga-DFO-c(RGDyK) (1–2 MBq/mouse, 1 μg DFO-c(RGDyK)). The animals were sacrificed by cervical dislocation at 30 and 90 min post-injection (p.i.). Organs and tissues of interest (blood, spleen, pancreas, stomach, intestines, kidneys, liver, heart, lung, muscle, bone, and tumor) were collected, weighed, and measured in the gamma counter. The results were expressed as percentage of injected dose per gram organ (% ID/g).

PET/CT imaging of the experimental animals was performed with an Albira PET/SPECT/CT small animal imaging system (Bruker Biospin Corporation, Woodbridge, CT, USA). The animals were r.o. injected with [^68^Ga]Ga-DFO-c(RGDyK) at a dose of 5–7 MBq corresponding to ~2 μg of DFO-c(RGDyK) **9** per animal. Anesthetized animals were placed in a prone position in the Albira system before the start of imaging. Static PET/CT images were acquired over 30 min, starting 30 and 90 min after injection for both healthy and tumor-bearing (U-87 MG and M21 vs. M21-L) mice. A 10-min PET scan (axial FOV 148 mm) was performed, followed by a double CT scan (axial FOV 110 mm, 45 kVp, 400 μA, at 400 projections). ThesScans were reconstructed with Albira software (Bruker Biospin Corporation, Woodbridge, CT, USA) using the maximum likelihood expectation maximization (MLEM) and filtered backprojection (FBP) algorithms. After reconstruction, the acquired data was viewed and analyzed with the appropriate software (PMOD software, PMOD Technologies Ltd., Zurich, Switzerland and VolView software, Kitware, Clifton Park, NY, USA).

## 4. Conclusions

We have developed synthetic protocols for simple preparation of RGD peptides conjugated with deferoxamine. Based on the combination of solid-phase and solution-phase synthesis (post-cleavage oligopeptide modification), the conjugation via lysin side chain was regioselectively achieved due to a high level of orthogonality in the structure of corresponding intermediates. Furthermore, with respect to high crude purities, only two purification steps within the reaction sequence were required. The protocols can be applied to the preparation of various analogical conjugates bearing lysine in a peptide moiety.

Moreover, here we report for the first time, to our best knowledge, a DFO-based RGD peptide for radiometal labeling. The deferoxamine-based c(RGDyK) conjugate **9** could be easily radiolabeled with Ga-68 with high radiochemical purity, thus not requiring further purification steps. [^68^Ga]Ga-DFO-c(RGDyK) showed excellent in vitro stability in human serum and PBS, high affinity for αvβ3 integrin, rapid predominantly renal elimination, and good tumor-to-background ratios, indicating that it may be applicable for imaging of tumors expressing αvβ3 integrins. Additionally, the use of DFO as a chelating moiety also allows labeling of the studied DFO-c(RGDyK) with different radiometals.

## Data Availability

The data presented in this study are available in the article or Appendix A. The raw datasets are available from the corresponding authors on reasonable request.

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
