# Peer review of "[^68^Ga]Ga-DFO-c(RGDyK): Synthesis and Evaluation of Its Potential for Tumor Imaging in Mice"

_ijms, 2021, doi:10.3390/ijms22147391_

Round 1

Reviewer 1 Report

The manuscript # ijms-1240751 submitted by Petrik M., et al. introduces a cyclic RGD analog, coupled to the acyclic chelator DFO for labeling with the diagnostic PET radionuclide Ga-68. Synthesis using in part a solid support is thoroughly described, followed by radiochemical and biological characterization in avb3-positive cells and rodents.

The subject harmonizes well with the requirements of the IJMS special issue. The paper is in general well written and the experimental work rather fair.

I am somewhat sceptic about the originality of the work and the impact it may have in the field of diagnostic radiopeptides. In view of so many previously reported RGD-based diagnostic radiopeptides for imaging tumor-related angiogenesis, on what grounds have the authors opted for yet another analog carrying this particular chelator? Some clear argumentation on this point would greatly enhance interest on the paper. Why DFO and why c(RGDyK) vs. existing versions? And have you met the expectations from these choices? How results from these choices compare with already published work? Such sober, critical and clear argumentation and overview is missing, despite a few faint comments scattered in the text.

The authors are asked to consider some minor points as well:

  • pg. 5, ln 138: ...120 min of incubation. Incubation where?
  • pg. 5, Table 1. How about in vivo stability? You have reported in pg. 2, lns-74-76, that introduction of D-amino acids and cyclization were previously shown to improve the in vivo stability of RGD analogs. How this relates to your compound?
  • Figure text on x/y axis: It would be very helpful to increase fonts and numbers, to help the reader. They are too small, especially in the BioD-diagrams. Figure 5: Why showing only coronal views of the animals? It would be excellent to have the 3D PET/CT images as well, similarly to Figure 3.
  • Pg. 12, 3.5 cell studies: Explain if the M21 cell line (avb3-positive) is a stably transfected one or not and if it is also commercially available. If it is transfected, are you using a selection antibiotic agent, such as geneticin.
  • The displacement study you are using cells incubated at 37oC with your (radio)ligand: How do you know you have reached the state of equilibrium and that you are not loosing either radioligand or ligand due to enzymatic breakdown, or internalization? If any of these processes occur, your calculations of IC50 are not reliable.
  • As radioligand you are using [68Ga]Ga-NODAGA-c(RGDyK): Is this a single chemical entity or a mixture of labeled and unlabeled bioconjugate? Do you have the Kd of your radioligand? Without the above conditions truly established, you cannot report an IC50.
  • The resulting displacement curve in Figure 2 shows a 50% nonspecific part. Such high non-specific values imply extensive sticking of your radioligand and cannot lead to a reliable calculation.
  • Pg. 13, 4. Conclusions: The most important conclusions on the originality and significance of this work are missing. You also mention that your radiotracer shows high selectivity for the avb3: Where do you base this conclusion? Have you performed binding assays for other type of integrins? I don't see any in the manuscript.

Author Response

Reviewer 1

The manuscript # ijms-1240751 submitted by Petrik M., et al. introduces a cyclic RGD analog, coupled to the acyclic chelator DFO for labeling with the diagnostic PET radionuclide Ga-68. Synthesis using in part a solid support is thoroughly described, followed by radiochemical and biological characterization in avb3-positive cells and rodents.

The subject harmonizes well with the requirements of the IJMS special issue. The paper is in general well written and the experimental work rather fair.

We thank the reviewer for positive evaluation of our manuscript.

I am somewhat sceptic about the originality of the work and the impact it may have in the field of diagnostic radiopeptides. In view of so many previously reported RGD-based diagnostic radiopeptides for imaging tumor-related angiogenesis, on what grounds have the authors opted for yet another analog carrying this particular chelator? Some clear argumentation on this point would greatly enhance interest on the paper. Why DFO and why c(RGDyK) vs. existing versions? And have you met the expectations from these choices? How results from these choices compare with already published work? Such sober, critical and clear argumentation and overview is missing, despite a few faint comments scattered in the text.

We agree with the reviewer that the topic related to RGD-based peptide radiodiagnostics is not new. However, we believe that novelty can be seen in the use of DFO as a chelator for radiolabeling of selected RGD peptide. We have decided to use DFO as it is a molecule that has been widely studied in the past for its ability to bind many different metal ions. DFO was approved for medical use by FDA already in 1968 under the brand name Desferal and has thus been in clinical use for a long time. Although DFO is well accessible and established chelator in nuclear medicine for peptide and especially antibody radiolabeling, to our knowledge, DFO-based RGD peptides were not studied yet. c(RGDyK) is considered as one of the most prominent structures for the development of molecular imaging compounds in order to determine αvβ3 expression. It was widely evaluated in preclinical research (e.g. Li et al., Bioconjugate Chem., 2007; Jeong et al., JNM, 2008; Oxboel et al., Int J Mol Imaging., 2012; Boros et al. Nucl Med Biol., 2012) and was also reported in clinical trials (e.g. Van der Gucht et al., Clin Nucl Med., 2016; Gnesin et al., EJNMMI Res., 2017; Durante et al., EJNMMI Res., 2020) related to radiolabeled RGD peptides for tumor imaging. The obtained results are compared towards previously published work in the Results and Discussion section of the manuscript. We have shown that [68Ga]Ga-DFO-c(RGDyK) can be used for imaging of tumors expressing αvβ3 integrins, thus our main expectations were met. The major benefit we see in the versatility of DFO, which should allow labeling of DFO-c(RGDyK) with different radionuclides, which could open ways for SPECT imaging and theranostic applications using the same ligand.

We have addressed some of these points in the Results/Discussion and Conclusions sections of the manuscript now.

The authors are asked to consider some minor points as well:

  • pg. 5, ln 138: ...120 min of incubation. Incubation where?

We have added the information regarding the incubation temperature now.

  • pg. 5, Table 1. How about in vivo stability? You have reported in pg. 2, lns-74-76, that introduction of D-amino acids and cyclization were previously shown to improve the in vivo stability of RGD analogs. How this relates to your compound?

We did not specifically tested the metabolic stability of the studied radiotracer in this study. However, certain indicator of radiotracer performance in vivo results already from the in vitro serum stability, which was in the case of [68Ga]Ga-DFO-c(RGDyK) excellent (>97 %). We also believe, that if there would be any major issue with the in vivo stability of [68Ga]Ga-DFO-c(RGDyK), we could not see the clear difference of the tracer uptake in the positive (M21 cell line) versus negative (M21-L cell line) tumor in tumor-bearing animals as indicated in figure 5B. Last but not least, the in vivo biodistribution of [68Ga]Ga-DFO-c(RGDyK) was in a full agreement with other similar 68Ga-labeled RGD peptides reported previously.

  • Figure text on x/y axis: It would be very helpful to increase fonts and numbers, to help the reader. They are too small, especially in the BioD-diagrams. Figure 5: Why showing only coronal views of the animals? It would be excellent to have the 3D PET/CT images as well, similarly to Figure 3.

We agree that increase the fonts will help the readers, however in the current size of the images/graphs, we cannot enlarge the labels substantially. We believe that if our manuscript is published, it will be just as online version and thus the readers can enlarge the figures with a zoom function.

We decided to use coronal slices for Figure 5 to better show the effectiveness of combined PET/CT imaging and correlation between PET and CT scans. The fused PET/CT image as coronal slice can perfectly visualize the uptake of the radiotracer in tumor tissue showing the clear correlation between PET and CT. On Figure 5B, CT scan shows both, positive and negative tumor, while the radioactive signal can be seen in the positive tumor only (except for the excretory organs, of course). We could not show this on 3D image under the similar setting as Figure 3 was created. The exact position of the tumors (mainly the negative one) would not be visible on 3D image as you can see it on the coronal slice image. We believe that the images of [68Ga]Ga-DFO-c(RGDyK) as coronal slices are in tumor-bearing animals more illustrative, but we are ready to provide also 3D images, if the reviewer insists. 

  • Pg. 12, 3.5 cell studies: Explain if the M21 cell line (avb3-positive) is a stably transfected one or not and if it is also commercially available. If it is transfected, are you using a selection antibiotic agent, such as geneticin.

M21 cell line was stably transfected with αvβ3 integrin receptor. To our knowledge, M21 cell line is not commercially available. We have obtained it as a kind gift from prof. Clemens Decristoforo (Medical University, Innsbruck, Austria) as it is stated in the manuscript.

The displacement study you are using cells incubated at 37oC with your (radio)ligand: How do you know you have reached the state of equilibrium and that you are not loosing either radioligand or ligand due to enzymatic breakdown, or internalization? If any of these processes occur, your calculations of IC50 are not reliable.

We have tested various incubation times from 30 min up to 90 min (see Figure 2A). The [68Ga]Ga-DFO-c(RGDyK) uptake is increasing during first 60 min, but it is remaining more or less the same between 60 min and 90 min. Based on this finding, we have set the incubation time to 60 minutes for the IC50 determination.

  • As radioligand you are using [68Ga]Ga-NODAGA-c(RGDyK): Is this a single chemical entity or a mixture of labeled and unlabeled bioconjugate? Do you have the Kd of your radioligand? Without the above conditions truly established, you cannot report an IC50.

The radioligand [68Ga]Ga-NODAGA-c(RGDyK) used for IC50 determination is not a single chemical entity, it is a mixture of “labeled” and “unlabeled (cold)” variant. We are aware that the gold standard used for this purpose, 125I-echistatin, would be a better choice, but unfortunately, this compound is not commercially available since 2016. Kapp et al., Sci Rep., 2017 published IC50 of c(RGDyK) as 3.8 ± 0.42 nM or Wei et al., Nucl Med Biol., 2009 referred IC50 of the same compound as 3.7 nM. We have calculated IC50 of our tracer as 2.35 ± 1.48 nM. Based on these similarities we believe that the IC50 value calculated for DFO-c(RGDyK) is not out of the reality. We did not determine the Kd value of our radioligand. The Kd value indication is not a standard parameter given in the relevant literature focused on the research of radiolabeled RGD-peptides. The IC50 values are presented by majority of the groups without disclosing Kd value for the radioligand.

  • The resulting displacement curve in Figure 2 shows a 50% nonspecific part. Such high non-specific values imply extensive sticking of your radioligand and cannot lead to a reliable calculation.

We are aware that our inhibition curve is not an ideal one. However, when we looked at the inhibition curves related to similar research in an appropriate articles published (e.g. Knetsch et al., EJNMMI., 2011), we can see also relatively high non-specific binding despite using classical radioligand for this purpose (125I-echistatin). As mentioned in previous comment, although it would be better to use e.g. 125I-echistatin as a radioligand for the displacement study and blocking did not work ideally, the results seem to be in an accordance with previously published data.

  • Pg. 13, 4. Conclusions: The most important conclusions on the originality and significance of this work are missing. You also mention that your radiotracer shows high selectivity for the avb3: Where do you base this conclusion? Have you performed binding assays for other type of integrins? I don't see any in the manuscript.

We have modified the conclusions according to the reviewer’s comments now.

We did not perform binding assays on other type of integrins. You are right that the wording including selectivity was not correct in this case. We have changed the sentence accordingly.

Reviewer 2 Report

This is a great study: well-designed, well-executed, with a clear description of the results. There are just a few issues to be sorted out:

line 65: neither -> either

line 69: 'tumor model'

line 76: 'as one of the most' should be replaced by 'to be'

line 77: 'standardly' should be replaced by 'routinely'

line 124: 'activity of up to'

line 127-134: should be transferred to the introduction or deleted

line 138: around should be replaced byapproximately

line 152: the inhibitor should be named

Fig. 2C: unit of the y axis is % per mg. Usually, displacement in a competition binding experiment is given either as CPM or % of the non-displaced tracer. Please consider revising.

line 504/505: The open data concept supported by the journal would go beyond presenting data in processed, aggregated form as in the current manuscript. Typically, this would involve publicly sharing direct experimental data, as they were measured in each replicate. Usually, this is best be done by copying the raw data into separate sheets of a excel file (one for each type of experiment, eg binding, LogP, stability, biodistribution etc). The authors should consider uploading this file to a public repository such as Zenodo or FigShare. Alternatively, they may provide this excel file as supplementary data.

Author Response

This is a great study: well-designed, well-executed, with a clear description of the results. There are just a few issues to be sorted out:

We thank the reviewer for a great evaluation of our manuscript.

line 65: neither -> either

line 69: 'tumor model'

line 76: 'as one of the most' should be replaced by 'to be'

line 77: 'standardly' should be replaced by 'routinely'

line 124: 'activity of up to'

line 127-134: should be transferred to the introduction or deleted

line 138: around should be replaced by approximately

line 152: the inhibitor should be named

Thank you for these valuable comments helping to improve the manuscript. We have incorporated majority of them into the new version of the article.

Fig. 2C: unit of the y axis is % per mg. Usually, displacement in a competition binding experiment is given either as CPM or % of the non-displaced tracer. Please consider revising.

We have decided for “%/mg of cellular protein” as it is more accurate than just “%” in our opinion. The uptake in % related to the amount of cells (cellular protein) eliminates differences in number of cells in the given wells of the 96-well plate.

line 504/505: The open data concept supported by the journal would go beyond presenting data in processed, aggregated form as in the current manuscript. Typically, this would involve publicly sharing direct experimental data, as they were measured in each replicate. Usually, this is best be done by copying the raw data into separate sheets of a excel file (one for each type of experiment, eg binding, LogP, stability, biodistribution etc). The authors should consider uploading this file to a public repository such as Zenodo or FigShare. Alternatively, they may provide this excel file as supplementary data.

We have modified the Data Availability Statement - The raw datasets are available from the corresponding authors on reasonable request.

Reviewer 3 Report

In this article authors have synthesized a gallium-68 labelled DFO-c(RGDyK) peptide conjugate and studied its in vitro and in vivo activity along with its tumor imaging potential. In my opinion the article is written well. Chemical characterization of compound is also satisfactory. I would like authors to add at least in vitro characterization of DFO-c(RGDyK) peptide conjugate 9 for comparison.

Author Response

In this article authors have synthesized a gallium-68 labelled DFO-c(RGDyK) peptide conjugate and studied its in vitro and in vivo activity along with its tumor imaging potential. In my opinion the article is written well. Chemical characterization of compound is also satisfactory. I would like authors to add at least in vitro characterization of DFO-c(RGDyK) peptide conjugate 9 for comparison.

We thank the reviewer for positive review on our manuscript.

We are not sure if we understand the reviewer comment correctly, but the same set of experiments as was used for [68Ga]Ga-DFO-c(RGDyK) in vitro characterization cannot be applied for non-radiolabeled compound. The obtained results of in vitro characterization of [68Ga]Ga-DFO-c(RGDyK) are based on measuring the radioactivity, what is not applicable for “cold” (non-radiolabeled) ligand.

Round 2

Reviewer 1 Report

see attached pdf

Author Response

The manuscript # ijms-1240751 submitted by Petrik M., et al. introduces a cyclic RGD analog,

coupled to the acyclic chelator DFO for labeling with the diagnostic PET radionuclide Ga-68.

Synthesis using in part a solid support is thoroughly described, followed by radiochemical and

biological characterization in avb3-positive cells and rodents. The subject harmonizes well with

the requirements of the IJMS special issue. The paper is in

general well written and the experimental work rather fair.

We thank the reviewer for positive evaluation of our manuscript.

I am somewhat sceptic about the originality of the work and the impact it may have in the field

of diagnostic radiopeptides. In view of so many previously reported RGD-based diagnostic

radiopeptides for imaging tumor-related angiogenesis, on what grounds have the authors opted

for yet another analog carrying this particular chelator? Some clear argumentation on this point

would greatly enhance interest on the paper. Why DFO and why c(RGDyK) vs. existing

versions? And have you met the expectations from these choices? How results from these

choices compare with already published work? Such sober, critical and clear argumentation and

overview is missing, despite a few faint comments scattered in the text.

We agree with the reviewer that the topic related to RGD-based peptide radiodiagnostics is not

new. However, we believe that novelty can be seen in the use of DFO as a chelator for

radiolabeling of selected RGD peptide. We have decided to use DFO as it is a molecule that

has been widely studied in the past for its ability to bind many different metal ions. DFO was

approved for medical use by FDA already in 1968 under the brand name Desferal and has thus

been in clinical use for a long time. Although DFO is well accessible and established chelator

in nuclear medicine for peptide and especially antibody radiolabeling, to our knowledge, DFObased

RGD peptides were not studied yet. c(RGDyK) is considered as one of the most

prominent structures for the development of molecular imaging compounds in order to

determine αvβ3 expression. It was widely evaluated in preclinical research (e.g. Li et al.,

Bioconjugate Chem., 2007; Jeong et al., JNM, 2008; Oxboel et al., Int J Mol Imaging., 2012;

Boros et al. Nucl Med Biol., 2012) and was also reported in clinical trials (e.g. Van der Gucht

et al., Clin Nucl Med., 2016; Gnesin et al., EJNMMI Res., 2017; Durante et al., EJNMMI Res.,

2020) related to radiolabeled RGD peptides for tumor imaging. The obtained results are

compared towards previously published work in the Results and Discussion section of the

manuscript. We have shown that [68Ga]Ga-DFO-c(RGDyK) can be used for imaging of tumors

expressing αvβ3 integrins, thus our main expectations were met. The major benefit we see in

the versatility of DFO, which should allow labeling of DFO-c(RGDyK) with different

radionuclides, which could open ways for SPECT imaging and theranostic applications using

the same ligand. We have addressed some of these points in the Results/Discussion and

Conclusions sections of the manuscript now.

It is good that this information is now added making it clearer to the reader why the authors

went for the DFO chelator.

Two more points: a) It would be best to define which radiometals are a good match for DFO,

as detailed in the references provided and b) in line 90, Ref.18 is not correct. It should be 17

(Ye et al.).

We have addressed these two reviewer’s points in the new version of the manuscript now.

The authors are asked to consider some minor points as well:

• pg. 5, ln 138: ...120 min of incubation. Incubation where?

We have added the information regarding the incubation temperature now.

Add the medium as well. Incubation wherein? In a solution, in plasma, in serum? Where?

The exact procedure is described in Materials and Methods section in detail. However, we have

included the information also to the Results and Discussion section as proposed by the reviewer.

• pg. 5, Table 1. How about in vivo stability? You have reported in pg. 2, lns-74-76, that

introduction of D-amino acids and cyclization were previously shown to improve the in vivo

stability of RGD analogs. How this relates to your compound?

We did not specifically tested the metabolic stability of the studied radiotracer in this study.

However, certain indicator of radiotracer performance in vivo results already from the in vitro

serum stability, which was in the case of [68Ga]Ga-DFO-c(RGDyK) excellent (>97 %). We

also believe, that if there would be any major issue with the in vivo stability of [68Ga]Ga-

DFOc(RGDyK), we could not see the clear difference of the tracer uptake in the positive (M21

cell line) versus negative (M21-L cell line) tumor in tumor-bearing animals as indicated in

figure 5B. Last but not least, the in vivo biodistribution of [68Ga]Ga-DFO-c(RGDyK) was in

a full agreement with other similar 68Ga-labeled RGD peptides reported previously.

One cannot make conclusions on in vivo stability by extrapolating results of in vitro studies.

Many enzymes are ecto-enzymes not present in the biological fluids. Furthermore, tumor

uptake doesn’t guarantee that the compound is in vivo stable.

Pls., rephrase accordingly and be accurate.

Yes, we agree with the reviewer that one cannot make conclusions on in vivo stability by

extrapolating results of in vitro studies. We did not make any conclusions on in vivo stability

of [68Ga]Ga-DFO-c(RGDyK) in the manuscript as we did not test in vivo stability of the

compound. We just discuss the in vitro stability of [68Ga]Ga-DFO-c(RGDyK) in the

manuscript, which was tested in different media. We also comment on in vivo behaviour of

[68Ga]Ga-DFO-c(RGDyK), which was in line with other reported radiolabelled monomeric

RGD-based peptide conjugates, however we do not comment on in vivo stability of the

compound as already mentioned above.

• Figure text on x/y axis: It would be very helpful to increase fonts and numbers, to help the

reader. They are too small, especially in the BioD-diagrams. Figure 5: Why showing only

coronal views of the animals? It would be excellent to have the 3D PET/CT images as well,

similarly to Figure 3.

We agree that increase the fonts will help the readers, however in the current size of the

images/graphs, we cannot enlarge the labels substantially. We believe that if our manuscript is

published, it will be just as online version and thus the readers can enlarge the figures with a

zoom function.

You can increase the fonts and even use a double line. Alternatively, you may remove

description within the Figure(s) and provide them in an eligible way in the legend. It is just not

nice for the reader.

We have enlarged the fonts and numbers. We hope that it is better readable now.

We decided to use coronal slices for Figure 5 to better show the effectiveness of combined

PET/CT imaging and correlation between PET and CT scans. The fused PET/CT image as

coronal slice can perfectly visualize the uptake of the radiotracer in tumor tissue showing the

clear correlation between PET and CT. On Figure 5B, CT scan shows both, positive and

negative tumor, while the radioactive signal can be seen in the positive tumor only (except for

the excretory organs, of course). We could not show this on 3D image under the similar setting

as Figure 3 was created. The exact position of the tumors (mainly the negative one) would not

be visible on 3D image as you can see it on the coronal slice image. We believe that the images

of [68Ga]Ga-DFO-c(RGDyK) as coronal slices are in tumor-bearing animals more illustrative,

but we are ready to provide also 3D images, if the reviewer insists.

I suggest to use both 3D and coronal views to have both an overall and specific aspect of your

compound’s performance.

We have added the 3D volume rendered images to Figure 5 and enlarged the font of the

descriptions.

• Pg. 12, 3.5 cell studies: Explain if the M21 cell line (avb3-positive) is a stably transfected one

or not and if it is also commercially available. If it is transfected, are you using a selection

antibiotic agent, such as geneticin.

M21 cell line was stably transfected with αvβ3 integrin receptor. To our knowledge, M21 cell

line is not commercially available. We have obtained it as a kind gift from prof. Clemens

Decristoforo (Medical University, Innsbruck, Austria) as it is stated in the manuscript.

Kindly reply also to the antibiotic question: Which antibiotic have you been using to keep the

transfected clones only alive?

We did not use any selection antibiotic during subculturing of M21or M21L cells, because these

cells are stable in the integrin expression and no selection antibiotic is needed. In addition, the

cells do not possess any gene coding the resistance to selection antibiotics. These cell lines were

developed by dr. Cheresh and dr. Spiro using multiple FACS sorting as described in the article

- Cheresh D.A and Spiro R.C., J Biol Chem, 1987.

Line 419: Replace (stably transfected with αvβ3 integrin-positive) by (transfected to stably

express αvβ3 integrin; αvβ3 integrin-positive)

We apologize for inaccuracy in the description of these cell lines. As it is described in the article

by Cheresh D.A and Spiro R.C., J Biol Chem, 1987, the M21 and M21L cells are not

transfected, but selected by FACS. We have rephrased the text in methodological section

accordingly.

The displacement study you are using cells incubated at 37oC with your (radio)ligand: How do

you know you have reached the state of equilibrium and that you are not loosing either

radioligand or ligand due to enzymatic breakdown, or internalization? If any of these processes

occur, your calculations of IC50 are not reliable.

We have tested various incubation times from 30 min up to 90 min (see Figure 2A). The

[68Ga]Ga-DFO-c(RGDyK) uptake is increasing during first 60 min, but it is remaining more

or less the same between 60 min and 90 min. Based on this finding, we have set the incubation

time to 60 minutes for the IC50 determination.

In Figure 2A you do not show or discriminate internalized from membrane-bound fraction and

consequently you do not know if internalization occurs. You also do not know that your

compound remains stable under these conditions. At 37oC enzymatic activity is high and also

internalization. This means that you do not have kinetic equilibrium and thus you may not use

the equations needed for calculating the IC50 values.

We are aware that we have not investigated the internalization of our radioligand. However, it

was previously described (e.g. Knetsch et al., NMB, 2015 or Zhai et al., MIB, 2016) that the

internalization of similar RGD-based conjugates is relatively low (cca 5%). We therefore

presume that internalization will not have high impact on the results of displacement study.

Concerning stability during 60 min of incubation, the IC50 study was performed in the fresh

buffer and not in the culture media. In the buffer, there is very low or more likely negligible

concentration of extracellular enzymes, which could break down our radioligand.

• As radioligand you are using [68Ga]Ga-NODAGA-c(RGDyK): Is this a single chemical entity

or a mixture of labeled and unlabeled bioconjugate? Do you have the Kd of your radioligand?

Without the above conditions truly established, you cannot report an IC50.

The radioligand [68Ga]Ga-NODAGA-c(RGDyK) used for IC50 determination is not a single

chemical entity, it is a mixture of “labeled” and “unlabeled (cold)” variant. We are aware that

the gold standard used for this purpose, 125I-echistatin, would be a better choice, but

unfortunately, this compound is not commercially available since 2016. Kapp et al., Sci Rep.,

2017 published IC50 of c(RGDyK) as 3.8 ± 0.42 nM or Wei et al., Nucl Med Biol., 2009

referred IC50 of the same compound as 3.7 nM. We have calculated IC50 of our tracer as 2.35

± 1.48 nM. Based on these similarities we believe that the IC50 value calculated for DFOc(

RGDyK) is not out of the reality. We did not determine the Kd value of our radioligand. The

Kd value indication is not a standard parameter given in the relevant literature focused on the

research of radiolabeled RGD-peptides. The IC50 values are presented by majority of the

groups without disclosing Kd value for the radioligand.

One is allowed to use a radioligand in a competition binding assay when it is a well established

one. Alternatively, one can characterize a new radioligand providing its Kd. Otherwise, one

cannot report a valid IC50 value for the competitor.

It looks like the fundamental principles are not followed.

We have already commented on that issue in the previous response and changed the text in the

manuscript accordingly now. The Kd value for very similar RGD-based conjugate (NOTAc(

RGDfK)) was published by Hedhli et al., Sci Rep, 2017 as 9.6 pM. This value is even lower

than Kd value published for 125I-echistatin (McLane et al., Biochem J, 1994), where the

authors determined the Kd value = 153 pM.

• The resulting displacement curve in Figure 2 shows a 50% nonspecific part. Such high nonspecific

values imply extensive sticking of your radioligand and cannot lead to a reliable

calculation.

We are aware that our inhibition curve is not an ideal one. However, when we looked at the

inhibition curves related to similar research in an appropriate articles published (e.g. Knetsch

et al., EJNMMI., 2011), we can see also relatively high non-specific binding despite using

classical radioligand for this purpose (125I-echistatin). As mentioned in previous comment,

although it would be better to use e.g. 125I-echistatin as a radioligand for the displacement

study and blocking did not work ideally, the results seem to be in an accordance with previously

published data.

Again, when you have a 50% non-specific binding, the equations do not apply. All these critical

points should be considered and discussed in the paper and not pretend that this is a real IC50

value of your compound: which compound in fact? Here you have a mixture!!!!

Pls., revise accordingly.

Yes, we have a mixture of labelled and non-labelled ligand in the IC50 study, but the stated

concentration deals with total concentration of the radioligand (not only the labelled fraction).

The calculation of IC50 is therefore not affected by the fact that there is a mixture of two ligands

(hot and cold one). Concerning high non-specific binding, we have revised the text in the

Results and Discussion section accordingly.

• Pg. 13, 4. Conclusions: The most important conclusions on the originality and significance of

this work are missing. You also mention that your radiotracer shows high selectivity for the

avb3: Where do you base this conclusion? Have you performed binding assays for other type

of integrins? I don't see any in the manuscript.

We have modified the conclusions according to the reviewer’s comments now. We did not

perform binding assays on other type of integrins. You are right that the wording including

selectivity was not correct in this case. We have changed the sentence accordingly.

Round 3

Reviewer 1 Report

I have carefully read the 3rd version of the ms. The authors have made a decent effort to address almost all comments.

I do believe that the ms has much improved by the changes undertaken, although I would prefer a more critical view of the methods applied and overall results.

For example, I would personally refrain from the last part of last phrase in the conclusions, in view of the fact that other chelators, like DOTA, have already provided this opportunity. Thus, the ways for SPECT imaging and theranostic applications have already been opened long ago:

"Additionally, the use of DFO as a chelating moiety allows also labeling of the studied DFO-c(RGDyK) with different radionuclides, which could open ways for SPECT imaging and theranostic applications."

Author Response

I have carefully read the 3rd version of the ms. The authors have made a decent effort to address almost all comments.

I do believe that the ms has much improved by the changes undertaken, although I would prefer a more critical view of the methods applied and overall results.

For example, I would personally refrain from the last part of last phrase in the conclusions, in view of the fact that other chelators, like DOTA, have already provided this opportunity. Thus, the ways for SPECT imaging and theranostic applications have already been opened long ago:

"Additionally, the use of DFO as a chelating moiety allows also labeling of the studied DFO-c(RGDyK) with different radionuclides, which could open ways for SPECT imaging and theranostic applications."

We would like to thank the reviewer for his critical review, which helped to improve our manuscript. We have modified the conclusions according to the reviewer's recommendations.